# On the Scaling Flaws of Verifier-Guided Beam Search in Mathematical Reasoning

## Abstract

Large language models (LLMs) struggle with multi-step mathematical reasoning, for which inference-time scaling—via sequential or parallel scaling—has emerged as a promising strategy. While recent advances have focused on sequential scaling, we revisit the less-explored parallel scaling approach, verifier-guided beam search, to examine its limitations. In this paper, we argue that its strength is, paradoxically, also its limitation: verifiers can boost performance under limited sample sizes by elevating promising reasoning paths, yet the same mechanism can also hide or cut off the valid paths that lead to correct answers. Empirically, we uncover a systematic issue–scaling flaws–in verifier-guided beam search, across models, benchmarks (GSM8K, MATH, AIME25), and verifier types (outcome value models, process reward models). Specifically, the search outperforms repeated sampling at small sample sizes but its advantage diminishes—and ultimately reverses—as the sample size grows. We attribute this to verifier failures: imperfect verifiers misrank candidates and can erroneously prune all valid paths, with these effects exacerbated on more challenging scenarios. To mitigate verifier failures, we explore reducing reliance on verifiers and conduct preliminary investigations using two simple methods. Overall, our findings expose fundamental limitations of verifier-guided beam search and explain why this line has struggled to realize its potential.

## 1 Introduction

Multi-step mathematical reasoning is challenging to LLMs (Hendrycks et al., 2021; Zheng et al., 2022). Recent studies have identified inference-time scaling, which scales computational efforts during the inference phase to achieve higher reasoning performance, involving sequential scaling (Jaech et al., 2024; Guo et al., 2025) and parallel scaling (Brown et al., 2024; Snell et al., 2024; Wu et al., 2024b), as a promising strategy to enhance performance. Parallel scaling explores multiple solution paths simultaneously via repeated sampling or tree search externally. Conversely, sequential scaling internalizes this exploration into a single, extended chain of thought, allowing for autonomous self-verification and self-correction. While sequential scaling (exemplified by O1 and DeepSeek-R1) has recently attracted substantial attention, it remains unclear why parallel scaling has struggled to keep pace. Parallel scaling begins with repeated sampling (Brown et al., 2024), which increases inference-time computation by generating multiple independent attempts from the model. This raises the chance that at least one attempt succeeds, enabling LLMs to solve more problems. Building on this insight, search-based approaches have emerged to guide computation toward more effective reasoning paths (Snell et al., 2024; Wu et al., 2024b).

Search reallocates computational resources by evaluating and selecting partial paths during generation. A common approach for path evaluation uses verifiers (Snell et al., 2024; Wu et al., 2024b), such as outcome value models (OVMs) (Yu et al., 2024) and process reward models (PRMs) (Lightman et al., 2024), to score and rank candidates, prioritizing valid paths. This makes verifier-guided beam search effective for challenging problems with sparse valid solutions, offering advantages over repeated sampling when the sample size is limited.

However, we observe that **verifier-guided beam search experiences diminishing advantages from scaling**. Its performance improves more slowly than repeated sampling, ultimately becoming less effective.

Through extensive experiments across multiple models, benchmarks (GSM8K, MATH, AIME25), and verifier types (OVMs, PRMs), we demonstrate that this phenomenon is systematic and robust. We refer to this phenomenon as *scaling flaws of verifier-guided beam search*.

We attribute the scaling flaws to *verifier failures*: imperfect verifiers frequently misrank candidate paths and can erroneously prune all valid solutions from the search space. This issue becomes more severe as the candidate pool grows, since valid paths become more widely distributed across problems while verifiers struggle to identify them. Moreover, we find that scaling flaws intensify on more challenging problems—precisely the scenarios where beam search is expected to provide the greatest benefits.

To mitigate these issues, we explore simple methods that reduce reliance on verifiers during beam search, such as stochastic selection and one-time Monte Carlo rollouts. Our preliminary results show that these approaches can partially alleviate verifier failures, suggesting promising directions for future work.

**Contributions** of this paper are three fold.

- This work identifies and analyzes the scaling flaws of verifier-guided beam search.

- We pinpoint verifier failures as the primary cause of these flaws.

- Our analysis reveals that these issues are more severe for challenging problems, raising concerns about the development of verifier-guided beam search algorithms and application in real-world settings.

## 2 Background: Verifier-Guided Beam Search

### 2.1 Problem Definition

This section begins by defining mathematical reasoning questions and introducing two widely employed solution frameworks: repeated sampling and beam search in Sec. 2.2.

**Definition. A mathematical reasoning question** $q$ *requires a step-by-step solution path* $S = [s^1, \ldots, s^T, a]$ *to be addressed, where* $s^i$ *represents the i-th step, $T$ is the number of steps, and $a$ is the final answer.*

Mathematical reasoning (Cobbe et al., 2021; Hendrycks et al., 2021) suffers from error propagation issues–errors in earlier steps affect later ones, compromising the final answer. Recent studies show that LLMs can address more challenging problems through repeated sampling (Brown et al., 2024).

**Repeated Sampling** LLMs can solve some challenging problems through multiple attempts (Cobbe et al., 2021; Brown et al., 2024), i.e. repeatedly sampling a set of solution paths $\left\{ S_k \right\}_{k=1}^{K}$ independently from the generator in parallel. Increasing the number of attempts $K$ often improves *coverage* (see Sec. 3.2). However, repeated sampling becomes inefficient for challenging problems, like competition-level mathematics problems (Hendrycks et al., 2021), where it often demands many more attempts to find a correct solution (Brown et al., 2024).

### 2.2 Beam Search

Repeated sampling generates multiple complete solutions independently. Although simple, it often requires many attempts to solve difficult problems. In contrast, general *search* strategies improve efficiency by exploring partial solutions incrementally, evaluating candidates at each step, and pruning unlikely paths early to prevent errors from propagating to full solutions. This paper focuses on *beam search* thanks to its simplicity and highly competitive performance [1].

*Beam search* intervenes in both the **generation** and **selection** stages at the step level, while exploring multiple paths in parallel.

---

[1] Moreover, although more advanced search methods such as Monte Carlo Tree Search (MCTS) (Wan et al., 2024; Snell et al., 2024; Chen et al., 2024) exist, prior studies have shown that MCTS does not consistently yield advantages in practice for this type of problem.

**Generation Stage** Given a question $q$, at each step $t$, the generator produces $K$ candidate partial paths

$$\mathbb{S}^{(1:t)} = \left\{ S_k^{(1:t)} \right\}_{k=1}^K,$$

where each candidate is defined as

$$S_k^{(1:t)} = [s_k^1, s_k^2, \ldots, s_k^t],$$

i.e., the $k$-th partial path up to step $t$.

**Selection Stage** A *scoring function* $f$ evaluates these candidates, assigning scores

$$\mathbb{V}^{(1:t)} = \left\{ v_k^{(1:t)} \right\}_{k=1}^K,$$

where $v_k^{(1:t)} = f\left( S_k^{(1:t)}; q \right)$ is the score of the $k$-th partial path. The candidates are then ranked by their scores, and the top $b$ paths are retained. Each of the top $b$ paths generates $\frac{K}{b}$ new candidates, maintaining a total of $K$ candidates for the next step.

This process repeats until all $b$ paths terminate, producing $b$ complete solution paths (see Algorithm 1). The hyperparameter $b$ controls the number of parallel paths. Increasing $b$ or $K$ allows the algorithm to explore a broader search space and improves its ability to solve more complex problems.

### 2.3 Verifier-guided Beam Search

Beam search usually use *verifiers* as the scoring functions $f$ (Yu et al., 2024; Chen et al., 2024; Snell et al., 2024). Verifiers (Lightman et al., 2024; Yu et al., 2024) are commonly employed as scoring functions to evaluate candidate, determining which paths to be further explored. In this work, we focus on the two most widely used types of verifiers–Outcome-supervised Value Models (Yu et al., 2024) and Process-supervised Reward Models (Lightman et al., 2024). See Appendix A.3 for the training procedure.

- **Outcome-Supervised Value Model (OVM)** The OVM (Yu et al., 2024) evaluates each candidate by estimating the probability of arriving at a correct answer from the given partial path, assigning higher scores to candidates with greater future potential. It assumes that each local step with the highest probability of success ultimately leads to the correct answer. We refer to beam search using OVM for evaluation as "OVM-guided search".

- **Process-Supervised Reward Model (PRM)** The PRM (Lightman et al., 2024) evaluates each candidate by predicting its step correctness, assigning higher scores to candidates with more correct preceding steps. It assumes that each correct local step guides to the correct final answer. We refer to beam search using PRM for evaluation as "PRM-guided search".

Verifiers play a key role in candidate evaluation and selection, directly influencing the search success. When they correctly identify valid paths, they can steer the search towards correct solutions more efficiently than repeated sampling. Interestingly, verifiers might misguide beam search and cause scaling issues, see Sec. 3.

## 3 A Pilot Study on Scaling Verifier-Guided Beam Search

In this section, we present extensive experiments and analysis on verifier-guided scaling beam search, showing that verifier-guided beam search suffers *scaling flaws*: it outperforms repeated sampling at small sample scale but fails to sustain this superiority, yielding inferior results at larger scales under the same number of sampled paths. Furthermore, expanding the candidate pool size fails to deliver performance gain.

### 3.1 Experimental Settings for Scaling

**Benchmarks** We perform experiments on three mathematical reasoning datasets: GSM8K (Cobbe et al., 2021), MATH (Hendrycks et al., 2021), and AIME25 [2]. The experiments are conducted under four distinct

---

[2]https://huggingface.co/datasets/math-ai/aime25

settings, including two in-distribution (GSM8K and MATH) and one out-of-distribution (OOD) scenarios (AIME25), as detailed in Appendix A.2.

**Models**   We use Mistral 7B (Jiang et al., 2023) and DeepSeekMath 7B (Shao et al., 2024) for GSM8K, and for MATH we use these two models plus Qwen2.5-Math 7B (Yang et al., 2024). For AIME25, we use Qwen2.5-Math 7B. For the in-distribution setting, the base models are trained on the corresponding training sets to serve as the generators. The OVMs used in each setting are initialized from these generators. For PRMs, we leverage the open-source Math-Shepherd dataset (Wang et al., 2024). The generators are first fine-tuned on a subset of this dataset to ensure that the training data of the generators and the PRMs share a similar distribution. The PRMs are then initialized from the corresponding generators and trained under supervision using process labels. For the AIME25, we use the models from the MATH setting. Qwen3-8B (Yang et al., 2025) is the latest model, but it produces long, sequential chains of thought (CoT) rather than the traditional CoT that our beam search algorithm operates on. [3]  We also configured it to produce traditional CoT, the results are shown in Appendix A.6.1. To further reinforce our analysis and findings, we incorporate the open-source Skywork-o1-Open-PRM-Qwen-2.5 7B, a state-of-the-art 7B PRM, as a strong verifier for exploration.

**Scaling Beam Search**   Search performance is driven by two factors: the number of parallel explored paths and the candidate pool size. While increasing either factor via higher computational budgets should intuitively enhance results, we systematically analyze their scaling laws. Specifically, we examine: (1) the number of parallel explored paths $b$, with $K/b$ fixed at 8, and (2) the number of generated candidates $K$, with $b$ fixed at 8. For the comparison between beam search and repeated sampling, we align them in terms of "sample size", which represents the number of sampled path generated by each algorithm. For beam search, the sample size corresponds to the number of parallel explored paths, $b$, while for repeated sampling, it corresponds to the number of attempts (See the explanation in Appendix A.4). Notably, under a fixed sample size, scaling the candidate size has no effect on repeated sampling, and its performance remains constant. Each experiment is repeated three times, and we report the average coverage (i.e. the fraction of problems for which at least one sampled path is correct) along with their standard deviation. See implementation details in Appendix A.5.

## 3.2   The Metric for Path-finding during Search: Coverage

We first present the primary metric, coverage, capturing the effectiveness of path-finding in discovering the solution set, and then motivate this choice.

**Definition. Coverage** [4] *measures the fraction of problems for which at least one sampled path is correct. For a given question $q$ with ground answer $a_*^q$, let the algorithm generate a set of solution paths $\mathbb{S}^q = \{S_k^q\}_{k=1}^K$, which induce the answer set $\mathbb{A}^q = \{a_k^q\}_{k=1}^K$. We say the question $q$ is passed if: $\text{pass}(q, a_*^q, \mathbb{A}^q) = \mathbb{I}(a_*^q \in \mathbb{A}^q)$. Over a test set $\mathcal{Q}^{test}$ of $N$ questions, **coverage** is calculated as:*

$$\text{coverage} = \frac{1}{N} \sum_{q \in \mathcal{Q}^{test}} \text{pass}(q, a_*^q, \mathbb{A}^q)$$

**Rationale behind Coverage**   Another well-known metric is *precision* (Brown et al., 2024). It measures the validity of a single selected answer from the solution set and is highly sensitive to the answer selection strategy applied after path-finding (e.g., reward models (Snell et al., 2024)). We adopt the *coverage* rather than the precision for two reasons: (1) Coverage upper-bounds precision and equals it when oracle selection is available (e.g., automatic theorem proving (Zheng et al., 2022), code generation (Chen et al., 2021)) (2) Focusing on coverage removes confounding from answer selection, isolating the path-finding process. Please refer to Section 6 for the discussion on precision.

---

[3]Whereas traditional CoT relies on externalizing solution exploration through multiple parallel sequences, long sequential CoT internalizes this exploratory process within a single, extended trajectory. This represents a fundamental distinction in the underlying search mechanism.

[4]Another commonly used term for coverage is pass@k. However, the sample size of beam search corresponds to the beam width $b$. To avoid potential confusion, we therefore use the term coverage instead.

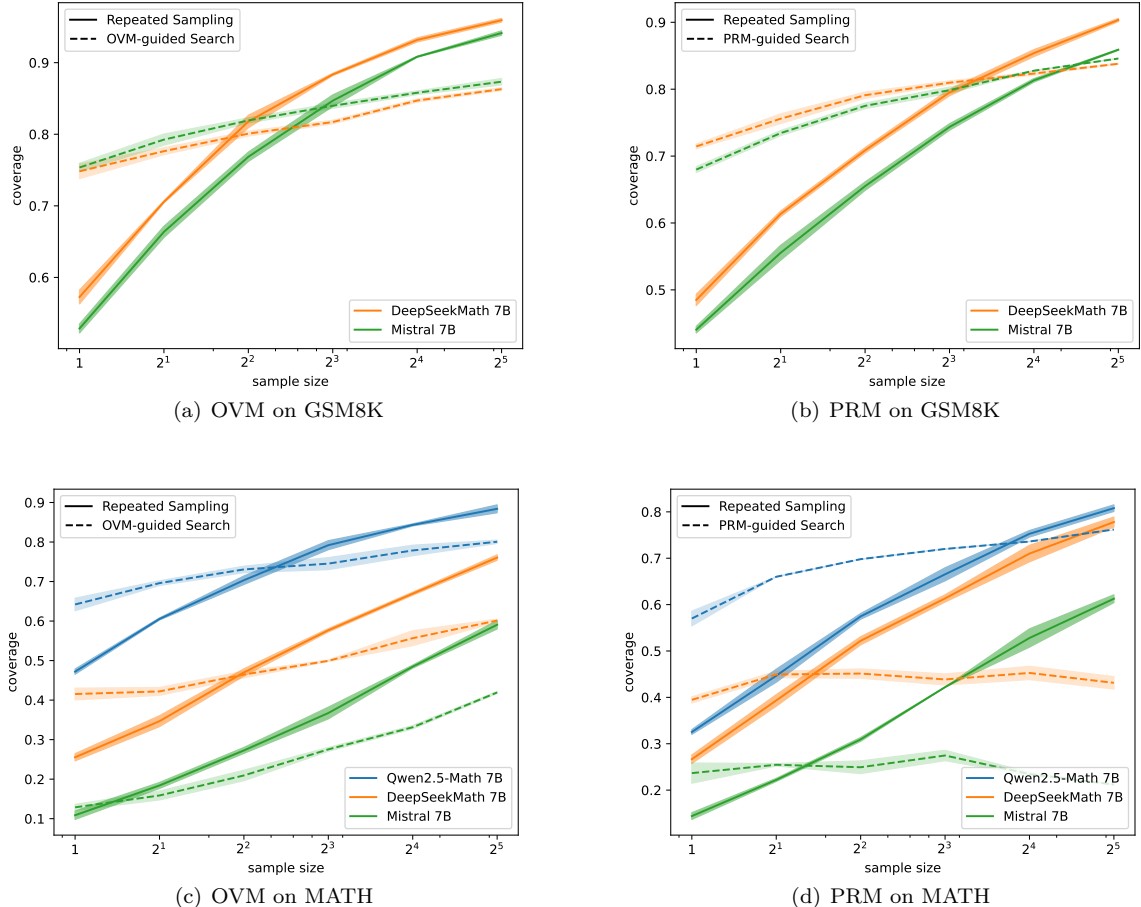

(a) OVM on GSM8K

(b) PRM on GSM8K

(c) OVM on MATH

(d) PRM on MATH

Figure 1: Scaling Flaws in OVM-guided search and PRM-guided search on GSM8K and MATH (increasing the number of sampled paths $b$, with $K/b$ fixed at 8). While verifier-guided beam search outperforms repeated sampling initially, its performance increases at a slower rate, ultimately underperforming repeated sampling.

### 3.3 Experimental Observation on Scaling Flaws

**Scaling Failed w.r.t. the Sample Size** $b$  When increasing the number of sampled paths $b$, as shown in Figure 1, both OVM-guided and PRM-guided search initially outperform repeated sampling; for example, by over 20% with PRM on MATH at $b = 1$. However, as $b$ increases, the performance gain of verifier-guided search diminishes and eventually reverses: by $b = 16$, PRM-guided search falls behind repeated sampling, and by $b = 32$, it underperforms by about 3%. Similarly, OVM-guided search is overtaken at $b = 8$ and lags by roughly 10% at $b = 32$. See the rationale of this setting in Appendix A.4.

**Scaling Failed w.r.t. the Candidate Size** $K$  Figure 2 shows the results for expanding the candidate pool size $K$. The results reveal that increasing the number of generated candidates does not improve and may even degrade the performance of verifier-guided beam search. Specifically, as the search candidate size grows, the coverage for both OVM and PRM-guided searches declines across all models. This suggests that the verifiers struggle to effectively rank and select the most promising paths when faced with a larger number of candidates, leading to a decrease in the likelihood of finding correct solutions. These findings underscore the challenges associated with scaling verifier-guided beam search and the need for more robust verifiers or alternative search strategies that can better handle large candidate sets.

**Exploration with a strong verifier**  The scaling flaws persists even with the state-of-the-art open-source PRM Skywork-o1-Open-PRM-Qwen-2.5 7B: When increasing the number of sampled paths, the coverage of PRM-guided search is eventually caught up and surpassed by repeated sampling, on both GSM8K and MATH.

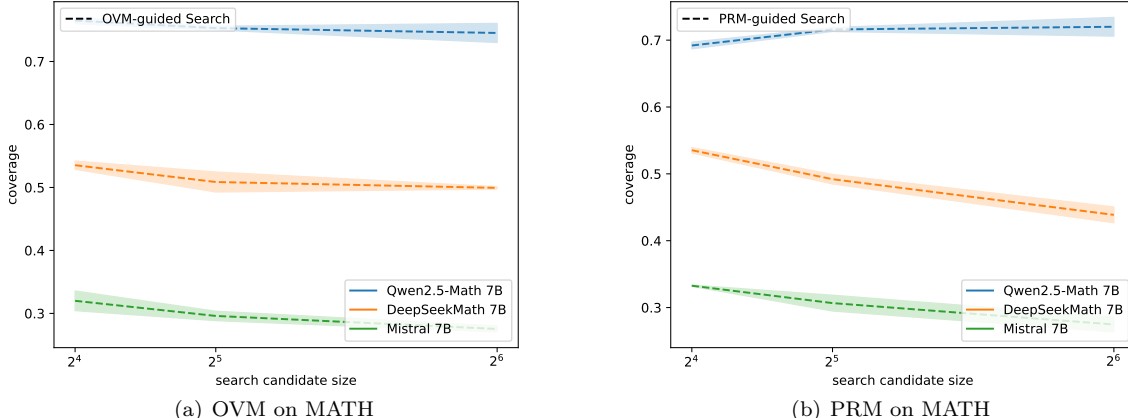

(a) OVM on MATH

(b) PRM on MATH

Figure 2: Scaling Flaws in OVM-guided search and PRM-guided search on MATH (expanding the candidate pool size $K$, with $b$ fixed at 8).

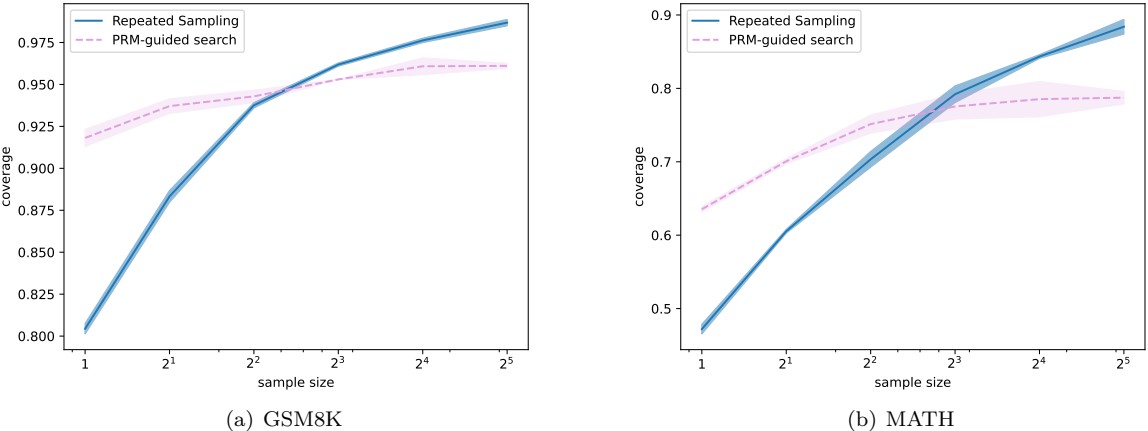

(a) GSM8K

(b) MATH

Figure 3: Scaling Flaws in PRM-guided search, using Qwen2.5-Math 7B as the generator and Skywork-o1-Open-PRM-Qwen-2.5 7B as the verifier (increasing the number of sample paths $b$, with $K/b$ fixed at 8).

This consistent reversal highlights a fundamental scaling flaw in verifier-guided beam search, which is formally described as below:

**Scaling Flaws of Verifier-guided Beam Search** : verifier-guided beam search experiences diminishing advantages as the sample or candidate size scales.

### 3.4 Scaling Flaws on More Challenging Problems

**Exacerbated Scaling Flaws on Challenging Problems** As shown in Figure 1 and Table 1, scaling flaws are more pronounced on MATH than on GSM8K. In Figure 1, we quantify performance degradation as the gap between beam search and repeated sampling when the number of sampled paths is set to 32. For OVM-guided search with DeepSeekMath and Mistral, the gap is approximately 10% on GSM8K, increasing to around 20% on MATH. Under PRM guidance, it rises from about 5%

Table 1: Increased average coverage of *verifier-guided beam search* over *repeated sampling* w.r.t. the number of sampled paths using Qwen2.5-Math 7B in AIME25. "Skywork" refers to the Skywork-o1-Open-PRM-Qwen-2.5 7B model.

| | #sample | AIME25 |
|---|---|---|
| OVM | 1 | 1.6% |
| | 32 | -24.4% |
| PRM | 1 | 1.4% |
| | 32 | -22.0% |
| Skywork | 1 | 4.9% |
| | 32 | -16.6% |

on GSM8K to nearly 30% on MATH. As shown in Table 1, the trend is further exacerbated on the more challenging dataset AIME25 – for the strongest Qwen2.5-Math model, the degradation expands from roughly 10% on MATH to approximately 24% on AIME25 under OVM guidance, and from nearly 3% on MATH to about 22% on AIME25 under PRM guidance. Even when using the state-of-the-art open-source PRM Skywork-o1-Open-PRM-Qwen-2.5 7B, the degradation remains substantial – up to 16.6%. If interested, please refer to Appendix A.6.2 for results on a more challenging dataset.

## 4 On the Cause of Scaling Flaws

To investigate the underlying causes of scaling flaws in verifier-guided beam search, this section presents an in-depth analysis. In Section 4.2, we attribute the search failures to primarily selection rather than generation, and reveal that incorrect selections are mainly due to imperfect verifiers – a phenomenon we term "verifier failures". In Section 4.3, we investigate the distribution of failed selection stages during the search, examining their correlation with the sparsity of candidate space. In Section 4.4, we analyze and demonstrate the severity of these verifier failures.

### 4.1 Experimental setup

In this section, we analyze the selection stages from two perspectives: (1) the first selection stage $\mathbb{S}^{(1:1)}$, with a large number of candidates ($K = 256$) to thoroughly examine the relationship between the number of candidates and the performance of verifier selection, covering both OVM selection and PRM selection; and (2) all selection stages $\mathbb{S}^{(1:1)}, \mathbb{S}^{(1:2)}, \ldots, \mathbb{S}^{(1:T)}$ during the OVM-guided search with $b = 8, K = 64$, since this configuration exhibits scaling flaws across benchmarks and models while maintaining an acceptable computational cost for valid path labeling. See Appendix A.6.4 for results with Skywork-o1-Open-PRM-Qwen-2.5 7B.

We consider a selection successful if, when valid candidates exist among the $K$ candidates, at least one of them is selected into the $b$ beams. A candidate is regarded as valid if it can lead to the correct final answer. To determine valid paths, we complete each partial path by rolling out multiple samples and verifying whether any of them successfully reach the correct answer. Specifically, we generate 4 rollouts per candidate for GSM8K and 16 for MATH. AIME25 follows the same experimental setting as MATH. See Appendix A.6.3 for the rationale behind the choice of the number of rollouts.

### 4.2 Verifier Failures Cause Search Scaling Flaws

Search failures can arise from either the generation stage or the selection stage—specifically, when no valid candidates are generated or when valid paths produced during generation fail to be selected.

**Generation vs. Selection Failures** *Search failures are largely attributable to selection failures.* We analyze all search processes in which problems fail to be solved and attribute these failures to either generation or selection. A failure is attributed to the generation stage if there is at least one intermediate step where no valid partial paths are generated. Conversely, it is attributed to the selection stage if, at any intermediate step, valid paths are produced but fail to be selected. As shown in Table 2, a large proportion (above 65%) of OVM-guided search failures occur during the selection stage, highlighting it as a critical issue. See the implementation details in Appendix A.5.4. [5]

Selection failures in verifier-guided beam search are directly attributable to verifiers. When verifiers fail to differentiate between valid and invalid paths, and mistakenly assign low ranks to all valid paths, none of them will be further explored, resulting in a selection failure. We refer to this issue as "verifier failures". Such failures, which prune all valid paths as failing to select any, ultimately lead to search failures.

To validate the role of verifier failures in contributing to search scaling flaws, we examine the relationship between the success of the selection stage and the number of candidates. Specifically, we analyze the

---

[5]We present only the results of OVM-guided search, as generation failures in PRM-guided search are expected to be similar due to the independence of the generation and selection stages.

Table 2: Fraction of OVM-guided search failure sources across benchmarks and models

| | GSM8K | | MATH | | | AIME25 |
| | DeepSeekMath | Mistral | Qwen2.5-Math | DeepSeekMath | Mistral | Qwen2.5-Math |
|---|---|---|---|---|---|---|
| Generation | 11.4% | 14.1% | 19.3% | 21.0% | 27.6% | 32% |
| Selection | 88.6% | 85.9% | 80.7% | 79.0% | 72.4% | 68% |

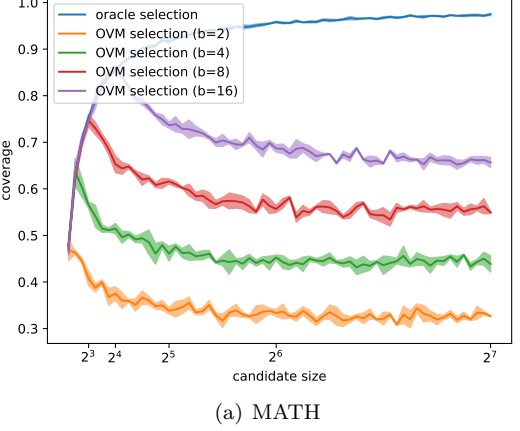

(a) MATH

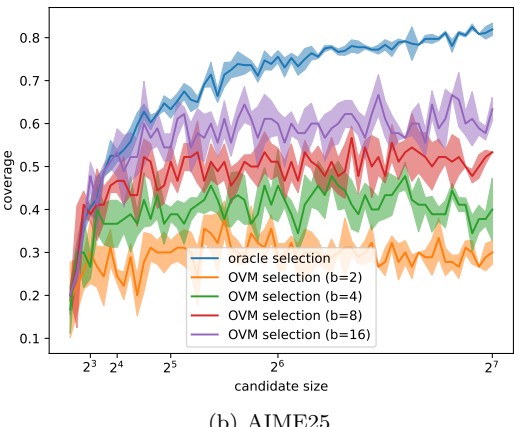

(b) AIME25

Figure 4: Scaling failures of OVM selection at the first selection stage across various beam sizes, as expanding the candidate pool size (Qwen2.5-Math 7B).

performance of verifier selection in correctly identifying and selecting at least one valid path as the number of candidates increases during the first selection stage. To ensure that the analysis accounts for the presence of valid paths in the candidate set, we use oracle selection performance as a baseline. This baseline serves as a reference for the maximum potential success of the selection process, independent of verifier performance.

**Verifier Selection Scaling Failures**   *There are verifier selection scaling failures during the selection stage.* As shown in Figure 4, verifier selection exhibits scaling failures. Specifically, as the candidate pool size expands, the performance of verifier selection improves only marginally, saturates, or even decreases, despite the presence of valid paths across more problems, as indicated by the oracle selection performance. This phenomenon is consistent across various beam sizes. While selecting and exploring more candidates improves robustness to verifier limitations—evidenced by the reduced gap between verifier selection and oracle selection performance—a significant gap persists even at the largest beam size tested, $b = 16$. These scaling failures suggest that verifier selection is a key bottleneck in the success of the selection process, and increasing the candidate size offers limited improvement in addressing this issue.

The scaling failure of verifier selection can explain the diminishing advantage of verifier-guided beam search. Initially, verifier-guided beam search is more efficient than repeated sampling, as it effectively selects valid paths and reallocates computational resources for several problems. However, as the candidate pool expands, even though valid paths are available across a broader range of problems, verifiers fail to identify and select them. In contrast, repeated sampling explores more paths without being constrained by verifier failures, ultimately outperforming verifier-guided beam search at larger scales.

### 4.3   Scenarios with More Challenging Problems

In this section, we analyze the failed selection stages during the search, showing that the search process is most hindered when valid paths are sparse.

**Sparser Candidate Space**    *Verifier selection failures occur and block search more often when valid paths are sparse.* We investigate the failed selection stages during the search and examine the valid path sparsity of these stages. Valid path sparsity is defined as the fraction of valid paths among the candidates. First, we group the valid path sparsity across all selection stages of unsolved problems into four uniform categories. Next, we identify the specific failure stage in each search process where verifier failures occur. We use these groupings to plot the distribution of valid path sparsity across the identified failure stages. See the implementation details in Appendix A.5.5.

As illustrated in Figure 5, the distribution of failed selection stages demonstrates a monotonic trend: as valid path sparsity decreases, the proportion of failed selection stages increases. This aligns with the intuition that identifying valid paths becomes more challenging in sparser candidate spaces, highlighting the verifier's limitations in handling such complexity. This finding reveals that verifier failures become increasingly significant when solving sparser correct solution spaces, amplifying the risk of search failure.

Although beam search is expected to offer greater efficiency than repeated sampling in solving more challenging problems by reallocating computational resources through effective selection, our observations suggest that these challenging scenarios are more susceptible to verifier failures, thereby exacerbating scaling flaws.

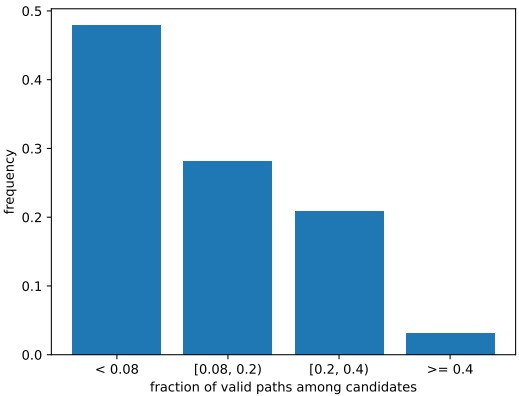

Figure 5: Distribution of OVM failures across groups based on valid path sparsity on MATH (Qwen2.5-Math 7B).

## 4.4   Verifier Failures Exceed the Tolerance of Beam Search

Verifier failures are identified as the primary cause of scaling flaws in the preceding analysis. This phenomenon is particularly striking as beam search is inherently designed to tolerate imperfect candidate evaluation.

Although practical verifiers inevitably assign inaccurate scores, the beam search algorithm naturally mitigates this risk by maintaining multiple parallel paths via the beam width ($b$). Consequently, a verifier does not need to perfectly score candidates or rank all valid paths above incorrect ones; it merely needs to retain *at least one* correct solution among the top-$b$ candidates. Nevertheless, our empirical analysis reveals that practical verifiers frequently fail to satisfy even this highly relaxed requirement, erroneously pruning all valid solutions from the candidate pool. The severity of these verifier failures manifests in three key aspects:

**Emergence under practical compute budgets**   Crucially, these verifier failures occur well within practical, rather than purely theoretical, computational limits. As illustrated in Figure 1, the performance curves of verifier-guided beam search and repeated sampling intersect at an early stage. Even our largest inference configuration ($b = 32, K = 256$) represents a realistic computational cost in practice, yet within this budget range, verifier-guided beam search already strictly underperforms repeated sampling.

**Failures in the simplest selection settings**   Even with a maximal retention ratio (e.g., $b/K = 0.5$), verifiers struggle to retain at least one correct path. For instance, as shown in Figure 4 (a), when the beam

width is $b = 16$ and the candidate size is $K = 32$, the performance gap between OVM-guided selection and oracle selection remains substantial, at approximately 15% on the MATH benchmark.

**Vulnerability of state-of-the-art verifiers**  We further observed that even the state-of-the-art open-source PRM falls short significantly under the simplest selection settings (Figure 8 (a)).

In summary, we empirically identify verifier failures as the primary cause of scaling flaws. Even state-of-the-art verifiers fail under the simplest selection scenarios, degrading search performance within realistic computational constraints. Furthermore, these failures are exacerbated on challenging problems, highlighting the fundamental difficulty of advancing verifier-guided beam search methods.

## 5 A Preliminary Exploration to Alleviate Verifier Failures

Imperfect verifiers can lead to verifier failures, obstructing the success of the search process. In this section, we observe the effectiveness of two simple methods at the selection stages for alleviating verifier failures, by reducing dependency on verifiers: stochastic selection and integration with one-time Monte Carlo rollout. See the implementation in Appendix A.5.6.

**Experimental Setup**  We evaluate these methods across all the selection stages of the search with $b = 8, K = 64$. For each method, we quantify performance loss as the decrease in the fraction of problems for which at least one valid path is selected, relative to the upper-bound performance attained by random selection.

**Stochastic Selection**  Imperfect verifiers can produce incorrect candidate rankings, potentially leading to misguided selection decisions. To mitigate the risk of over-reliance on erroneous rankings, we introduce stochasticity into the selection stage. Rather than deterministically selecting candidates based solely on verifier-predicted score rankings, we apply a softmax function to the candidates' scores and sample from the resulting probability distribution. This approach maintains a preference for high-scoring candidates while still allowing lower-scoring ones a chance to be selected, thereby reducing the risk of incorrectly pruning misranked valid paths.

We experiment with three temperature settings (0.01, 1, and 100). Higher temperatures reduce reliance on verifier evaluations, being closer to uniform selection. Conversely, lower temperatures increase dependence on verifier evaluations, approximating deterministic selection.

As shown in Table 3, stochastic selection improves the selection stage across all benchmarks. Interestingly, on GSM8K, a moderate temperature performs best by balancing verifier guidance and stochasticity. In contrast, higher temperatures are more beneficial for MATH settings. This pattern aligns with intuition: since MATH settings experience more severe verifier failures than GSM8K, reducing reliance on the verifier through increased randomness (i.e., higher temperature) is likely more advantageous in these scenarios.

**One-time Monte Carlo Rollout**  This method enhances candidate evaluation by incorporating simulated rewards with verifier-predicted scores. Specifically, we perform a one-time rollout for each partial path $S^{(1:t)}$ until completion and obtain the reward of the resulting full path. [6] We then linearly combine this reward $r$ with the verifier-predicted score $v^{(1:t)}$ using the formula $\lambda r + (1 - \lambda)v^{(1:t)}$, where $\lambda$ controls the balance between the simulated reward and the verifier's evaluation.

As shown in Table 3, increasing $\lambda$ generally leads to better results. Notably, the best result is achieved when relying entirely on the simulated reward, without incorporating the verifier-predicted score. [7] This underscores the limitations of verifiers in candidate evaluation.

---

[6] The reward is estimated by the same verifier based on the complete path.

[7] Crucially, while $\lambda = 1$ MC rollout bypasses verifier-predicted scores, it remains a deliberate search process that is fundamentally distinct from blind repeated sampling. Unlike the latter, which lacks any selection mechanism, the MC-based approach actively prunes the search space by using empirical future rewards as a selection heuristic.

Table 3: Results of performance loss with OVM baseline and variations of temperature and lambda. Lower is better.

| | GSM8K | | MATH | | | Average |
|---|---|---|---|---|---|---|
| | DeepSeekMath | Mistral | Qwen2.5-Math | DeepSeekMath | Mistral | |
| OVM | | | | | | |
| | 2.3% | 0.3% | 17.2% | 5.9% | 21.4% | 10.0% |
| *temperature* in stochastic selection | | | | | | |
| 0.01 | 1.9% | 0.3% | 17.4% | 6.5% | 20.1% | 9.6% |
| 1 | 0.0% | -0.5% | 1.3% | 0.7% | -0.2% | 0.3% |
| 100 | 0.1% | 0.1% | 0.0% | 0.4% | -0.4% | 0.04% |
| *lambda* for one-time Monte Carlo rollout | | | | | | |
| 0.5 | 0.5% | -0.5% | 16.4% | 6.7% | 21.1% | 8.8% |
| 0.75 | 0.4% | -0.5% | 14.8% | 6.2% | 18.4% | 7.5% |
| 1 | 0.0% | -0.6% | 6.7% | 4.2% | 15.3% | 5.1% |

## 6 Discussion on Precision

We have identified scaling flaws in verifier-guided beam search and analyzed that verifier failures bottleneck upper-bound performance (as measured by coverage). In this section, we shift to the other evaluation metric: precision. Specifically, we compare verifier-guided beam search against repeated sampling by evaluating the precision achieved when an answer selection mechanism is applied following the path-finding process.

**Experimental Setup**   We perform answer selection on the set of sampled paths generated by under the largest inference configuration (i.e. $b = 32, K = 256$). Two answer selection verifier settings are considered: (1) using the same PRM verifier that guides the search, and (2) employing a more powerful independent verifier, Skywork-o1-Open-PRM-Qwen-2.5 7B (denoted as Skywork). Experiments are conducted with Mistral, DeepSeekMath, Qwen2.5-Math on MATH, under PRM-guided beam search.

Table 4: The comparison of coverage and precision on MATH across models, under the largest inference configuration (i.e. $b = 32, K = 256$).

| | Repeated Sampling | | | PRM-Guided Search | | |
|---|---|---|---|---|---|---|
| | Coverage | Precision | Precision (Skywork) | Coverage | Precision | Precision (Skywork) |
| Mistral | 61.3% ± 0.8% | 17.9% ± 8.9% | 39.7% ± 7.5% | 21.2% ± 0.0% | 16.2% ± 0.0% | 19.6% ± 0.4% |
| DeepSeekMath | 77.8% ± 1.1% | 31.1% ± 10.1% | 58.1% ± 7.6% | 43.1% ± 1.3% | 33.6% ± 0.6% | 40.7% ± 1.2% |
| Qwen2.5-Math | 80.8% ± 0.5% | 43.3% ± 10.0% | 61.5% ± 6.8% | 76.2% ± 0.7% | 57.6% ± 0.5% | 64.4% ± 0.3% |

As shown in Table 4, we can find that

- *Coverage*: Repeated sampling achieves higher coverage than PRM-guided search across Mistral, DeepSeekMath, and Qwen2.5-Math.

- *Precision with the same verifier*: When following the setup of Brown et al. (2024), i.e., using the same verifier for both beam search and post-hoc answer selection, the precision of search becomes higher, outperforming repeated sampling for DeepSeekMath and Qwen2.5-Math, consistent with the findings in Brown et al. (2024). However, for the weakest model, Mistral, the precision of search remains lower than that of repeated sampling, mirroring the pattern observed in coverage.

- *Precision with a more powerful verifier*: When a stronger verifier is used for answer selection, the precision of repeated sampling surpasses that of search for Mistral and DeepSeekMath. For Qwen2.5-Math, search shows slightly higher precision, but the gap is reduced from 14.3% to 2.9%. This suggests that employing a stronger answer selection strategy can make repeated sampling superior in

precision. With an oracle verifier, precision would be identical to coverage, implying that repeated sampling would outperform beam search.

Overall, these results indicate that precision is highly sensitive to the effectiveness of the answer selection strategy. Hence, the apparent superiority of verifier-guided beam search reported in prior work (e.g., Brown et al., 2024) may partly stem from the limitations of the applied verifier in answer selection.

This observation underscores the significance of our findings regarding scaling flaws based on coverage; by abstracting away the confounding effects of answer selection, coverage provides a more robust upper bound on achievable performance and reveals the inherent limitations of the beam search paradigm.

## 7 Conclusion

Based on extensive empirical analysis across multiple models, benchmarks, and verifier types, this work identifies and explains the scaling flaws of verifier-guided beam search: while it outperforms repeated sampling at small sample sizes, its advantage diminishes and eventually reverses as computational scale increases. The core issue lies in verifier failures: imperfect verifiers frequently misrank and prune valid reasoning paths, especially in challenging problems where solution paths are sparse. This inherent limitation undermines the beam search's ability to benefit from increased compute, revealing a fundamental bottleneck in the approach. These findings help explain why verifier-guided beam search has struggled to match the recent success of sequential scaling methods like DeepSeek-R1, and suggest that future advances in mathematical reasoning may need to either significantly improve verifier reliability or explore alternative scaling paradigms that avoid verifier-dependent path selection altogether.

**Limitations** This study is limited to mathematical reasoning benchmarks, leaving the transferability of the observed scaling flaws to other domains unevaluated; only two simple selection stage strategies are explored, while deeper training-time or architectural fixes remain untouched; and all experiments employ standard beam search, so the findings may not extend to alternative parallel scaling schemes such as MCTS.

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

# A Appendix

Table 5: Summary of Notations Used in the Paper

| Notation | Description |
|---|---|
| $q$ | Mathematical reasoning question requiring a sequence of steps |
| $S$ | Solution path for a question, $S = [s^1, \ldots, s^T, a]$ |
| $s^i$ | $i$-th step in a solution path |
| $a$ | Final answer in a solution path |
| $T$ | Number of steps in a solution path |
| $y$ | Binary label (0 or 1) indicating the correctness of $a$ |
| $S^{(1:t)}$ | Partial solution path up to step $t$, $S^{(1:t)} = [s^1, \ldots, s^t]$ |
| $\mathbb{S}^{(1:t)}$ | Set of candidate partial paths $\mathbb{S}^{(1:t)} = \{S_k^{(1:t)}\}_{k=1}^K$ |
| $v^{(1:t)}$ | The score for the partial path $S^{(1:t)}$ |
| $\mathbb{V}^{(1:t)}$ | Set of scores for candidate partial paths $\mathbb{V}^{(1:t)} = \{v_k^{(1:t)}\}_{k=1}^K$ |
| $K$ | Number of candidates |
| $b$ | Beam size |

## A.1 Related Works

**Search algorithms**  Search algorithms often face a tradeoff between effectiveness and efficiency. Approaches like MCTS (Hao et al., 2023; Tian et al., 2024) improve effectiveness by incorporating backtracking, but at the cost of efficiency. Other methods prioritize efficiency with minimal sacrifice in effectiveness (Wu et al., 2024a). In this work, we use a simple beam search algorithm (Yu et al., 2024; Chen et al., 2024) for our experiments, focusing on highlighting challenges in the candidate evaluation and selection stage, orthogonal to these advanced techniques.

**Candidate evaluation in search**  Candidate evaluation is a crucial stage that determines which paths are more valuable for further selection and exploration. Some methods rely on the some rule-based heuristics (Xin et al., 2024), with limited effectiveness. Some approaches involve lookahead techniques to assess candidates by simulating their subsequent outcomes (Snell et al., 2024; Wan et al., 2024), which significantly increases computational cost. Other methods incorporate external verifier models (Yu et al., 2024; Snell et al., 2024) to evaluate each candidate. In this work, we focus on the challenges and limitations of this approach.

**Limitations of imperfect verifiers**    Gao et al. (2023) shows that imperfect verifiers are susceptible to reward hacking, particularly as the sample size increases. Specifically, in the coding domain, Stroebl et al. (2024) demonstrates that scaling through repeated sampling yields limited improvements due to the fallibility of test cases or verifiers; In the context of mathematical reasoning, Cinquin et al. (2025) investigates the limits of PRM-guided tree search in limited settings, finding that search does not surpass repeated sampling under precision. In our work, we identify verifier failures as the core cause of scaling flaws in verifier-guided search, extending the investigation of imperfect verifiers' impact from repeated sampling to the broader search paradigm.

## A.2    Benchmark and Model Settings

There are three benchmark settings, including two in-distribution and one OOD:

- **GSM8K**: The official training split is used for training, and the model is evaluated on the test split.

- **MATH**: The official training split, comprising 7,500 problems, is used for training, while evaluation is performed on the MATH500 (Lightman et al., 2024).

- **AIME25**: We use the MATH-trained models and evaluate on the AIME25 dataset.

Four main model backbones involved in this paper are Mistral-7B-v0.1 [8], deepseek-math-7b-base [9], Qwen2.5-Math-7B [10], and Qwen3-8B [11].

## A.3    Verifier Training

**OVM training dataset construction**    OVMs are trained on automatically constructed datasets, where the correctness of the final answer serves as the label for each instance. The training dataset is constructed from the generator and the given question-answer pairs: For each pair $(q, a) \in \mathcal{Q}$, the generator produces $n$ solution paths, resulting in $|\mathcal{Q}| \times n$ question-solution pairs. The label $y \in \{0, 1\}$ for each solution $S$ is determined by checking the correctness of the final answer, e.g. matching it to the ground truth $a$, with 1 indicating "correct" and 0 indicating "incorrect". This process generates a training dataset of $(q, S, y)$ tuples for value models.

**PRM training dataset**    PRMs are trained at a fine-grained step level, requiring annotations of step correctness. The label for each solution path $S$ is a sequence $\mathbf{y} = y_1 \cdots y_T$ $(\forall y_i \in \{0, 1\})$, where $y_i$ indicates the correctness of the $i$-th step. In this study, we use the open-source Math-Shepherd process data [12] Wang et al. (2024) to train the PRMs.

**Training losses**    Both OVMs and PRMs are trained with mean squared losses, with target labels representing distinct underlying semantics.

$$L^{\mathrm{OVM}}(q, S, y) = \sum_{t=1}^{T}(\mathrm{OVM}(q, S^{(1:t)}) - y)^2$$

$$L^{\mathrm{PRM}}(q, S, \mathbf{y}) = \sum_{t=1}^{T}(\mathrm{PRM}(q, S^{(1:t)}) - y_t)^2$$

---

[8]https://huggingface.co/mistralai/Mistral-7B-v0.1 (Apache-2.0 license)

[9]https://huggingface.co/deepseek-ai/deepseek-math-7b-base (Deepseek license)

[10]https://huggingface.co/Qwen/Qwen2.5-Math-7B

[11]https://huggingface.co/Qwen/Qwen3-8B

[12]https://huggingface.co/datasets/peiyi9979/Math-Shepherd

### A.4 Beam Search

The algorithm is shown in Algorithm 1.

When scaling sample sizes, we compare verifier-guided beam search and repeated sampling by aligning the beam width $b$. We deliberately adopt this setup, rather than comparing FLOPs or tokens generated, for the following reasons.

- *First, this setup enables a fair comparison at the selection level*, allowing us to directly examine the effectiveness of verifier selection. In beam search, the beam width $b$ indicates the number of paths explored in parallel. Although $K$ candidates are generated and considered at each iteration, only $b$ of them are selected and expanded, while others are pruned. Aligning $b$ therefore allows us to match the number of parallel explored paths rather than the computational cost. Moreover, repeated sampling can be viewed as a special case of beam search where $K$ candidates are implicitly generated and $b$ of them are randomly selected. Thus, verifier-guided beam search differs only in that the selection is guided by a verifier rather than random choice. Failures in this setup therefore highlight weaknesses in verifier scoring, selection, and pruning — i.e., the verifier-guided search underperforms even when it is allowed to guide selection strategically.

- *Second, this setup further underscores the limitations of verifier-guided search.* When aligning the beam width $b$, verifier-guided beam search actually requires more computation than repeated sampling, as it samples more (i.e. $K$) candidates. However, despite the increased computation, its performance is still inferior to repeated sampling. This contrast emphasizes the fundamental limitations of verifier-guided beam search.

---

**Algorithm 1** Beam Search

---

    **Input:** Question $q$, Beam size $b$, Sampled steps per state $K$, Maximum step count $T^{max}$
    **Output:** $b$ solution sequences for $q$
    **Model:** Generator and VM
 1: Initialize step sequences $\mathbb{S} \leftarrow \{\}$
 2: Sample initial steps $\{s_1^1, \ldots, s_K^1\}$
 3: Select $b$ steps via SELECTION($q$, $\{s_1^1, \ldots, s_K^1\}$, $b$, VM) and add to $\mathbb{S}$
 4: $t \leftarrow 1$
 5: **while** sequences in $\mathbb{S}$ are not complete and $t < T^{max}$ **do**
 6:    $\mathbb{S}_{\text{new}} \leftarrow \{\}$
 7:    **for** each sequence $S^{(1:t)}$ in $\mathbb{S}$ **do**
 8:        **for** $i = 1$ to $K/b$ **do**
 9:            $S_i^{(1:t+1)} = \text{Generator}(S_i^{(1:t)}; q)$
10:            $\mathbb{S}_{\text{new}} \leftarrow \mathbb{S}_{\text{new}} + S_i^{(1:t+1)}$
11:        **end for**
12:    **end for**
13:    $\mathbb{S}_{\text{new}} \leftarrow \text{SELECTION}(q, \mathbb{S}_{\text{new}}, b, \text{VM})$
14:    $\mathbb{S} \leftarrow \mathbb{S}_{\text{new}}$
15:    $t \leftarrow t + 1$
16: **end while**
    **return** $\mathbb{S}$

---

Although the methods in Section 5 demonstrate similarity to MCTS algorithms, there exist some important differences. For instance, UCT, the common selection strategy in MCTS, leverages visit statistics to balance exploration and exploitation, which extends far beyond simply introducing randomness and rolling out instances.

### A.5 Implementation Details

#### A.5.1 OVMs

**Training generators**   We train the base models (Mistral 7B, DeepSeekMath 7B, Qwen2.5-Math 7B, or Qwen3 8B) on the training sets of each setting. In MATH, we split the steps using period and newline characters. We normalize datasets to use the newline character as the marker for the end of each step across all tasks. In all datasets, supervised fine-tuning is performed for 2 epochs with a batch size of 128. We use a linear learning rate scheduler with a maximum learning rate of 2e-6 for Mistral 7B, 5e-5 for DeepSeekMath 7B, 2e-5 for Qwen2.5-Math 7B, and 1e-5 for Qwen3 8B. The AdamW optimizer (Loshchilov & Hutter, 2019) is used for training.

**Building training dataset for OVMs**   The dataset construction process is introduced in Appendix A.3. We sample 50 solution paths per problem in GSM8K, and 50 solution paths per problem in MATH. For GSM8K, we follow the setup in Yu et al. (2024), with a decoding temperature of 0.7 and top-k set to 50 for dataset collection. The maximum new token length is set to 400 for GSM8K. In MATH, we use a decoding temperature of 1, top-p of 0.98, and a maximum new token length of 2000. We apply vllm (Kwon et al., 2023) to accelerate the generation process.

**Training OVMs**   OVMs are initialized from the corresponding generator checkpoints and trained for one epoch, using the same backbone learning rate scheduler as the generator training. The batch size is set to 128 in GSM8K and to 512 in MATH. The optimizer used for training is AdamW.

#### A.5.2 PRMs

We use the open-source Math-Shepherd dataset Wang et al. (2024) to train both the generators and PRMs.

**Data extraction**   We extract training problems for each setting. Specifically, for the GSM8K task, we extract all problems from the training split of GSM8K, and for the MATH task, we extract all problems from the training split of MATH.

**Data preprocessing**   Since the data format in Math-Shepherd is inconsistent, we normalize the solution paths. We detect steps in each path, normalize them to be split by a newline character, and summarize the final answer in the format of "The answer is xx". For MATH problems, the final answer is enclosed in "\boxed{}".

**Training generators**   For each setting, we randomly select one correct solution for each training problem. If no correct solution is provided, we randomly select one other solution. The training parameters, including the number of epochs, learning schedule, batch size, and optimizer for each base model (Mistral 7B, DeepSeekMath 7B, Qwen2.5-Math 7B, or Qwen3 8B), are the same as those in Appendix A.5.1.

**Training PRMs**   We use all solution paths and annotations provided in Math-Shepherd to train PRMs, which are initialized from the corresponding generator checkpoints and trained for one epoch, using the same backbone learning rates as OVM and generator training. The batch size is set to 128 and AdamW is used for training.

#### A.5.3 Beam search

In GSM8K, we set the decoding temperature to 0.7, top-k to 50, maximum new token length to 400, and maximum number of steps to 10. In MATH, we set the decoding temperature to 1.0, top-p to 0.98, maximum new token length to 2000, and maximum number of steps to 30. In AIME25, we set the decoding temperature to 1.0, top-p to 0.98, maximum new token length to 4096, and maximum number of steps to 50. During the beam search process, we prioritize selecting non-duplicate steps. We use vllm in MATH to accelerate token sequence generation.

### A.5.4 Analysis of search failures

We calculate the fractions of generation and selection failures among all search failures as follows:

(1) Identify search failures: A search fails on question $q$ if $\text{pass}(q, a_*^q, \mathbb{A}^q) = 0$. Let $m$ be the number of such failed cases.

(2) Locate the first failed stage: For each failed search, we find the first stage where all $b$ selected sequences are invalid.

(3) Classify failures: If no valid path exists in the candidate set, it's a generation failure. Otherwise, valid paths exist but none are selected, it's a selection failure.

(4) Report: Fractions are computed over the total $m$ failed cases.

### A.5.5 Analysis of valid path sparsity

Under the coverage measurement, a problem is considered solved if any of the generated solution paths in the output set is correct; otherwise, it is unsolved. For solved problems, all generation and selection stages succeed, therefore producing at least one correct path. In contrast, each unsolved problem has at least one failed stage $t$ that hinders the search. For such problems, we collect the first failed stage $\mathbb{S}^{1:t}$ and all preceding successful stages $\mathbb{S}^{1:1}, \ldots, \mathbb{S}^{1:t-1}$.

Using all these stages from all unsolved problems, we compute the valid path sparsity for each stage $\mathbb{S}$ and group the results into four uniform bins, where "uniform" means that each bin contains an equal number of stages. We then assign all the first failed stages $\mathbb{S}^{1:t}$ into these bins and analyze their distribution. This enables us to examine the distribution of valid path sparsity, using the stages from the same problems as the baseline.

### A.5.6 Alleviating verifier failures

We explore to alleviate verifier failures at the selection stages, i.e. by modifying the implementation of Selection in Line 13 of Algorithm 1.

**Standard verifier selection**. Given the candidate set $\mathbb{S}^{(1:t)}$ and the verifier VM, we compute the verifier-predicted score for each candidate $v_k = f(S_k^{(1:t)}; q), \forall S_k \in \mathbb{S}^{(1:t)}$. Then, we select the top-$b$ candidates with the highest scores as follows

$$\left\{ S_k^{(1:t)} \,\middle|\, k \in \underset{k=1,\cdots,K}{\text{argtop}_b} \, v_k \right\}$$

**Stochastic selection**. Instead of greedily selecting the candidates with the highest scores, we sample from the softmax-normalized score distribution of these candidates. Specifically, we first compute the probability for each candidate as $p_k = \frac{\exp\left(f(S_k^{(1:t)}; q)/\tau\right)}{\sum_{j=1}^{K} \exp\left(f(S_j^{(1:t)}; q)/\tau\right)}$, where $\tau$ is a temperature parameter. Then, we select $b$ candidates according to this distribution:

$$\left\{ S_k^{(1:t)} \,\middle|\, k \sim \text{Categorical}(p_1, p_2, \ldots, p_K), \text{ sample } b \text{ times without replacement} \right\}.$$

**One-time Monte Carlo rollout.** In this method, we incorporate simulated rewards to enhance selection. Specifically, we first perform a one-time simulation to roll out each partial candidate to a complete solution:

$$S_k = \text{Generator}(S_k^{(1:t)}; q), \quad \forall S_k^{(1:t)} \in \mathbb{S}^{(1:t)}.$$

Then, we compute the reward for each simulated complete path: $r_k = f(S_k; q), \forall k$. Finally, we select the top-$b$ candidates with the highest combined scores, where $\lambda$ controls the balance between the simulated

reward and the verifier's evaluation:

$$\left\{ S_k^{(1:t)} \,\middle|\, k \in \operatorname*{argtop}_b_{k=1,\cdots,K} \lambda r_k + (1-\lambda) v_k \right\}$$

### A.6    More Experimental Results

#### A.6.1    Scaling flaws when using Qwen3 8B

As shown in Figure 6, Qwen3 also suffers from scaling flaws, and it does not demonstrate significant effectiveness over Qwen2.5-Math under this setting.

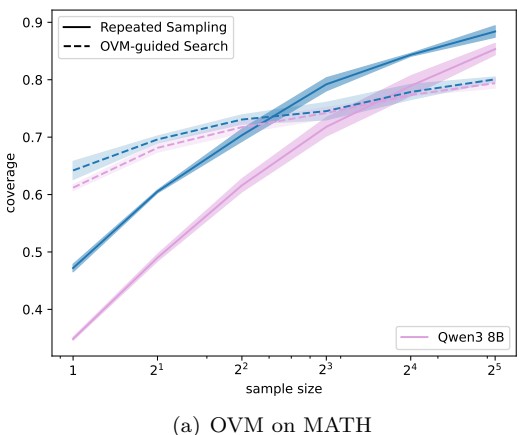
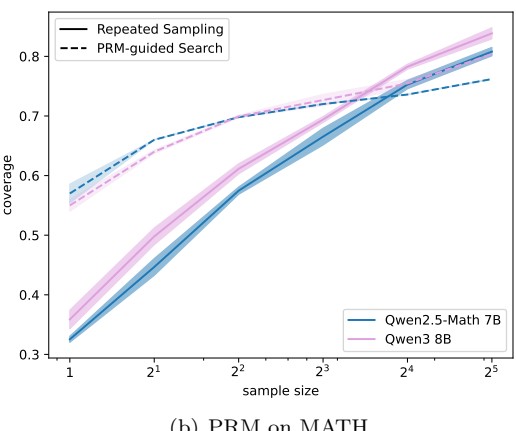

(a) OVM on MATH                                        (b) PRM on MATH

Figure 6: Scaling Flaws in OVM-guided search and PRM-guided search on MATH (increasing the number of sampled paths) using Qwen3 8B.

#### A.6.2    Scaling flaws on MinervaMath

Under the enhanced setting of using the state-of-the-art PRM Skywork-o1-Open-PRM-Qwen-2.5 7B , we extend our evaluation to the challenging dataset MinervaMath [13] in Table 6.

Table 6:    Average coverage when using Skywork-o1-Open-PRM-Qwen-2.5 7B to guide the search on MinervaMath (Qwen2.5-Math-7B, with $K/b$ fixed at 8).

| sample size | Repeated Sampling | PRM-Guided Search |
|---|---|---|
| 1 | $18.1\% \pm 1.4\%$ | $22.9\% \pm 1.7\%$ |
| 32 | $74.0\% \pm 0.8\%$ | $32.1\% \pm 1.5\%$ |

As shown in Table 6, although PRM-guided search initially outperforms repeated sampling by 4.8%, the gain vanishes and reverses into a substantial loss, with performance falling behind repeated sampling by 41.9%.

#### A.6.3    Analysis of the number of rollouts

We adopt a different number of rollouts across datasets mainly due to their varying difficulty levels. We use fewer rollouts for GSM8K because it is considerably easier than MATH. The actual numbers (4 for GSM8K and 16 for MATH) are chosen to balance effectiveness and computational cost. During the analysis of search failures and valid path sparsity distribution, we need to roll out all candidate paths at every iteration, which is computationally expensive. For example, in MATH with a search depth of 15 and a setting of $b = 8, K = 64,$

---

[13]https://huggingface.co/datasets/math-ai/minervamath

the total number of rollouts for a problem is $15 * 64 * 16 = 15360$. With 500 test problems, this amounts to approximately 7.68 million rollouts in total.

To verify that our chosen rollout numbers are reasonable, we plot the average sparsity of the first stages with respect to the number of rollouts on GSM8K and MATH, both with $K = 256$, in Figure 7.

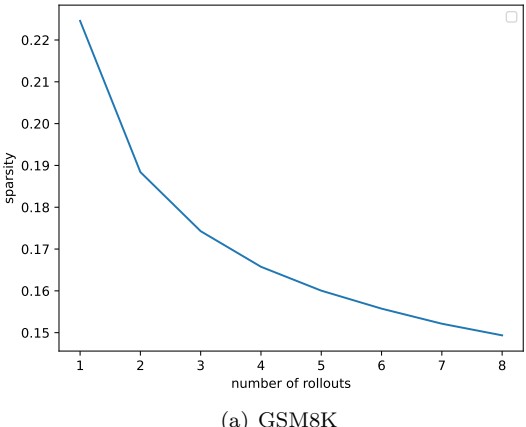
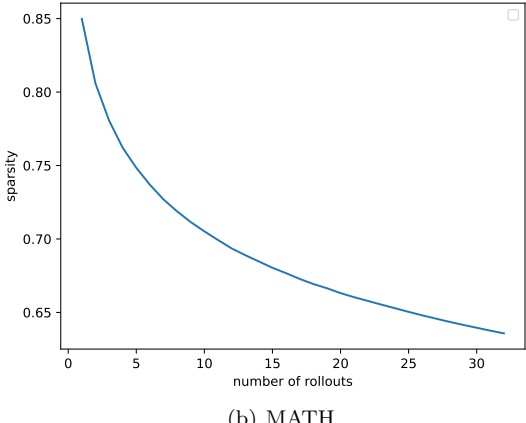

(a) GSM8K    (b) MATH

Figure 7: The average sparsity of the first stages w.r.t. the number of rollouts (Qwen2.5-Math 7B, with $K = 256$).

As shown in Figure 7, although the curves do not completely plateau, the rate of change slows significantly. Specifically, increasing from 4 to 8 rollouts on GSM8K changes only 1.6% of candidate decisions (approximately 4 out of 256), and increasing from 16 to 32 rollouts on MATH changes only 4.1%. Notably, this analysis focuses on the first stage, which requires the most rollouts to verify path validity. Therefore, for later stages, the potential bias introduced by the rollout number is even smaller.

This rollout-based labeling process can be conservative: a valid partial path might be mislabeled as invalid under the limited completions. However, we emphasize that this bias leads to an underestimation of selection failures. If a false negative occurs (a valid path is mislabeled as invalid), a failure that should have been attributed to the verifier's inability to select it is instead misattributed to the generator's inability to produce it. Consequently, the actual impact of selection failures is likely even more severe than the results reported in Table 2. This bias further reinforces our core claim that the verifier's selection capability—rather than the generator's capacity—is the primary bottleneck causing scaling flaws.

### A.6.4   Analysis of verifier failures when using Skywork-o1-Open-PRM-Qwen-2.5 7B

We examine the scaling of the strong verifier selection (Skywork-o1-Open-PRM-Qwen-2.5 7B) during the selection stage in Figure 8.

As illustrated in Figure 8, despite employing this state-of-the-art PRM, verifier selection still faces scaling failures – performance plateaus or deteriorates as candidate size increases, leading to a larger discrepancy relative to oracle selection. This suggests that Skywork-o1-Open-PRM-Qwen-2.5 7B, despite being state-of-the-art, still acts as a bottleneck in the search process.

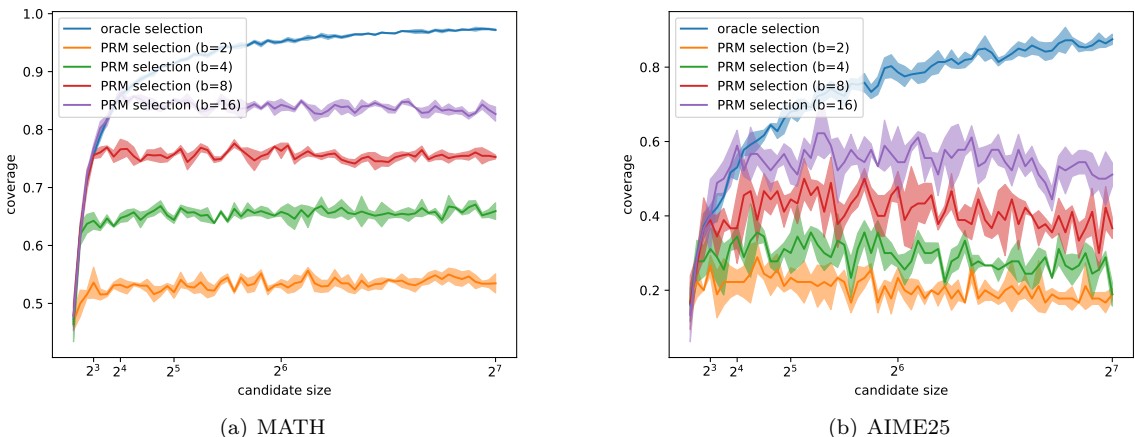

(a) MATH

(b) AIME25

Figure 8: Scaling failures of PRM selection at the first selection stage across various beam sizes, as expanding the candidate pool size (Qwen2.5-Math 7B, with Skywork-o1-Open-PRM-Qwen-2.5 7B as the PRM verifier).

