# OpenReview forum: "On the Scaling Flaws of Verifier-Guided Beam Search in Mathematical Reasoning"
_TMLR — Under review for TMLR_

### Review · Reviewer_dYSc · 2026-06-01

**Summary Of Contributions:**

This paper investigates a systematic limitation of verifier-guided beam search for LLM mathematical reasoning. The authors compare the coverage (fraction of problems for which at least one sampled path is correct) of repeated sampling against beam search augmented with a process reward model (PRM) or outcome value model (OVM). Their central finding (termed "scaling flaws") is that verifier-guided beam search initially outperforms repeated sampling at small sample sizes, but its advantage diminishes and eventually reverses as the sampling budget grows. This phenomenon is demonstrated across three LLMs, three benchmarks of increasing difficulty (GSM8K, MATH, AIME25), and two verifier types (OVM, PRM). The authors attribute this failure to imperfect verifiers that misrank candidates and erroneously prune all valid reasoning paths, a problem that worsens on harder problems where valid paths are more seldom. Two lightweight mitigations are explored: stochastic selection (softmax-temperature sampling over verifier scores) and one-time Monte Carlo rollouts.

**Strengths:**
- The core finding is robust across models, verifier types, and benchmarks of varying difficulty, lending credibility to the main claim.
- The analysis cleanly separates generation failures from selection failures (Table 2), with the latter dominating, which is a useful diagnostic contribution.
- The finding is practically relevant.

**Weaknesses:**
- Overall paper clarity is poor: key terms such as "sequential scaling", "parallel scaling", and "repeated sampling" are either undefined or only defined late in the paper.
- The paper frames "scaling flaws" as a novel discovery, but the finding is closely related to Stroebl et al. (2024), who demonstrate analogous degradation under imperfect verifiers in coding. The paper also does not discuss Cinquin et al., "Limits of PRM-Guided Tree Search for Mathematical Reasoning with LLMs", which investigates closely related questions and reaches similar conclusions. The difference over previous work is not sufficiently articulated.
- The comparison is limited to beam search versus repeated sampling, omitting common baselines such as majority voting, best-of-N with various aggregation functions, and Monte Carlo tree search.
- The proposed mitigations (Section 5) are weak: the best-performing fix (λ=1 Monte Carlo rollout) ignores the verifier entirely, which reduces to a form of repeated sampling rather than a genuine improvement to verifier-guided search.
- Results are averaged over only 3 random seeds, which is little for LLM reasoning evaluation and estimating standard deviations. Please consider using at least 5.

**Audience:**

Yes

**Audience Explanation:**

Inference-time scaling is a practically important research area. The TMLR community would benefit from the knowing the failure mode identified in the paper. Even if the finding is partly anticipated by Stroebl et al. (2024) and Cinquin et al. (2025), the systematic multi-model, multi-benchmark, multi-verifier demonstration adds value.

**Broader Impact Concerns:**

None. The paper studies a limitation of an existing method for LLM mathematical reasoning and poses no obvious ethical concerns.

**Claims And Evidence:**

Yes

**Claims Explanation:**

The main empirical claim (verifier-guided beam search exhibits a systematic performance reversal relative to repeated sampling) is convincingly demonstrated.

However, I have two concerns:

- **Validity labeling reliability** The generation-vs-selection decomposition in Table 2 depends on rollout-based identification of valid paths at each search step. This is standard practice in the PRM literature, but the specific use here is more demanding: a false negative (a valid path mislabeled as invalid due to insufficient rollouts) would misattribute a selection failure as a generation failure. The authors' own Figure 7b shows the sparsity estimate still declining at 16 rollouts on MATH, making the approximation error non-negligible. A brief acknowledgment of the directional bias this introduces into the attribution numbers would strengthen the analysis.
- **Coverage as the primary metric** The exclusive focus on coverage obscures a more nuanced picture. The precision results in Appendix A.6.3 (Table 6) show that verifier-guided search can retain advantages under some conditions (results for repeated sampling show much higher variance as PRM-guided search). This is currently buried in an appendix and deserves more prominent treatment in the main text.

**Requested Changes:**

**Critical for acceptance:**

1. **Improve clarity and motivation throughout.** The paper does a poor job motivating the problem and the background methods. The introduction jumps directly into the empirical setup without establishing why verifier-guided beam search was believed to be a promising direction in the first place, nor what the practical consequences of its failure are. The related work discussion is almost entirely absent from the main paper and a single appendix section is insufficient given the density of relevant prior work in inference-time scaling, reward model reliability, and test-time compute optimization. The authors should move and substantially expand the related work into the main text, covering at minimum: the current state of LLMs for mathematical reasoning, the landscape of test-time scaling strategies, and prior studies on the limitations of imperfect verifiers.
2. **Differentiate from Stroebl et al. (2024) and Cinquin et al.** The current related work (Appendix A.1) mentions Stroebl et al. in a single sentence and omits Cinquin et al. entirely. Given that both papers make closely related claims, the authors must provide an explicit comparison in the main text: what does this paper show that prior work does not?
3. **Promote and reconcile the precision results.** Table 6 should be moved or summarized in the main body, with a discussion of when verifier-guided search retains practical advantages under precision. The current framing risks overstating the negative result.
4. **Clarify the Monte Carlo rollout mitigation.** The best result (λ=1) ignores the verifier entirely. The authors should clarify whether this collapses to a form of repeated sampling with rollouts, and if so, what the computational overhead is relative to plain repeated sampling. Without this, the constructive contribution of Section 5 is unclear.
5. **Acknowledge validity labeling bias.** The rollout-based valid-path labeling used for the generation-vs-selection decomposition introduces a directional bias (underestimating valid paths, inflating generation failure rates). A brief acknowledgment and sensitivity analysis would strengthen the causal claims in Section 4.

**Would strengthen the work:**

6. **Extend the baseline comparison.** The paper compares only beam search and repeated sampling, omitting majority voting, best-of-N with various aggregation functions, and MCTS. At minimum, a qualitative discussion of whether the identified failure mode is expected to apply to MCTS.
7. **Increase the number of random seeds.** Three seeds is on the low end for LLM evaluation. While the trends appear visually robust, increasing to at least 5 seeds would strengthen confidence in the reported standard deviations.
8. **More precise use of the term "scaling".** The word "scaling" is used throughout the paper in several distinct senses (increasing the number of sampled paths, expanding the candidate pool, and increasing overall compute) without a clear definition being provided upfront. Replacing it with more descriptive language where possible, or providing a brief taxonomy of what is being scaled in each context at the start of the paper, would meaningfully improve clarity and precision.
9. **Harmonize the model selection across datasets in Table 2.** The authors use different subsets of models for GSM8K, MATH, and AIME25 in Table 2 without justification. Using a consistent set of models across all three benchmarks would make the results more directly comparable and strengthen the empirical narrative around how scaling flaws intensify with problem difficulty.
10. **Fix presentation issues:**
    - Share the y-axis across subplots in Figure 1 to facilitate direct comparison between OVM and PRM results.
    - Improve figure captions to be self-contained.
    - Fix Table 1 which overflow the margins.
    - Move the "Examination with a stronger verifier" paragraph into the models section where it belongs.
    - Spell out the G and S abbreviations in Table 2.
11. **Report inference time** alongside coverage to give a full picture of the computational cost tradeoffs between beam search and repeated sampling.

---

> ### Author Response · Authors · 2026-06-30
> **Rebuttal**
>
> Dear Reviewer dYSc,
>
> We sincerely thank you for your thoughtful evaluation, constructive feedback, and the time dedicated to reviewing our manuscript. We are highly encouraged that you find our main empirical claim—that verifier-guided beam search exhibits a systematic performance reversal relative to repeated sampling—to be "convincingly demonstrated" across multiple models, verifier types, and benchmarks of varying difficulty, and practically relevant to the TMLR community. We have addressed you critiques in the revised manuscript, and all corresponding updates are highlighted in blue.
>
> 1. **Regarding the concern on the clarity**
>
> Following you suggestions, we have comprehensively revised the manuscript to enhance the clarity of our paper. Specifically, we have explicitly defined key terms—including "sequential scaling", "parallel scaling", and "repeated sampling"—early in the Introduction, colored in blue and red. Furthermore, to make the usage of "scaling" clearer, we now explicitly specify and highlight the context for each experiment, carefully distinguishing between scaling the number of sampled paths and expanding the candidate pool. We have replaced the term of "sample size" with "the number of sampled paths" throughout the key results, which is more descriptive.
>
>
> 2. **Regarding discussion on closely related works**
>
> We sincerely appreciate the reviewer for highlighting these two relevant studies. We acknowledge that our work shares a high-level motivation with Stroebl et al. (2024) and Cinquin et al. (2025) regarding the limitations of verifiers. However, we maintain that our work provides a unique and necessary perspective. To clarify our contribution, we delineate the fundamental differences between our work and these studies as follows:
>
> (1) ***Comparison with Stroebl et al. (2024)***
>
> - *Nature of Verifier Failure*: The performance degradation observed by Stroebl et al. (2024) in coding stems primarily from incomplete test cases (a limitation of rule-based, ground-truth evaluation). In contrast, our study focuses on the intrinsic fallibility of model-based verifiers (e.g., reward models), which presents a more generalized challenge for LLM reasoning.
>
> - *Search Mechanism*: Their work primarily identifies the limits of unstructured repeated sampling, whereas our research investigates the dynamics of verifier-guided beam search, specifically examining how verifiers influence the pruning and selection process across reasoning steps.
>
> - *Core Implications*: Stroebl et al. (2024) argue that current performance gains are often spurious due to misleading metrics in coding (passing partial tests but failing overall). Conversely, we demonstrate that even with valid metrics in math, the theoretical upper bound of verifier-guided beam search is fundamentally constrained by verifier quality, leading to the "scaling flaws".
>
> (2) ***Comparison with Cinquin et al. (2025)***
>
> - *Breadth of Scaling Analysis*: Cinquin et al. (2025) primarily evaluate search performance at fixed scales to show that PRM-guided search is comparable but less efficient than repeated sampling. In contrast, our work provides a more comprehensive scaling landscape by systematically varying both the number of parallel paths and candidate pool sizes. This allows us to reveal a dynamic shift—specifically, how verifier-guided beam search transitions from outperforming repeated sampling with few number of sampled paths to underperforming it at larger scales.
>
> - *Evaluation Metric and Robustness*: Their study concludes that PRM-guided search does not surpass repeated sampling based on precision across various aggregation methods. Our work, however, shifts the focus to coverage (upper-bound performance). We demonstrate that the inherent limitations of verifier-guided search—our "scaling flaws"—persist regardless of the aggregation strategy, thereby revealing a more fundamental bottleneck in inference-time scaling.
>
> - *Generality across Models and Tasks*: While Cinquin et al. (2025) rely on a single generator-verifier pair, our study establishes the robustness of these findings by evaluating a diverse suite of model types, verifier types, and a wider range of benchmarks. This ensures that the identified "scaling flaws" are a systemic characteristic of current verifier-guided beam search rather than an artifact of a specific model configuration.

---

> ### Author Response · Authors · 2026-06-30
> **Rebuttal2**
>
> 3. **Regarding the concern on lacking of various aggregation methods**
>
> Indeed, various answer aggregation strategies (e.g., a reward model) can be applied after repeated sampling, in which the key evaluation metric becomes precision. The effectiveness of this answer aggregation strategy has a major influence on the results. For example, with an oracle answer selection mechanism, the precision would be exactly equal to the coverage. In practice, how well coverage translates to precision depends on how accurately the answer aggregation strategy can identify the correct answer from the candidate set. In our work, we focus on the upper-bound performance captured by the coverage, which isolates the effectiveness of specific answer aggregation strategies.
>
> 4. **Regarding the question on the best-performing Monte Carlo rollout setting**
>
> We clarify that while λ=1 MC rollout and repeated sampling both bypass intermediate verifier scores, they are fundamentally different in their decision-making mechanisms:
> - Repeated sampling is a "blind" process that samples complete paths randomly without intermediate evaluation or selection.
> - In contrast, λ=1 MC rollout remains a deliberate search process. It evaluates partial paths by simulating their potential success through full-path completions, using the empirical final rewards as a heuristic to prune the search space.
>
> In other words, λ=1 MC rollout replaces the predicted step-level scores (which we identify as a source of scaling flaws) with sampled future rewards to guide selection, whereas repeated sampling lacks any such selection mechanism.
>
> 5. **Regarding the concern on the randomness**
>
> We agree that more repetitions can reduce statistical noise. However, we limited our experiments to three repetitions for the following reasons:
>
> - The experiments are computationally expensive. For example, one OVM-guided run with 32 sample size in MATH takes ~1.75 hours on 4 A100-80GB GPUs. Our computational resources are limited.
> - Our focus is on scaling trends rather than absolute performance. Given the small standard deviations and clear trends, we consider three runs sufficient.
>
> To further verify robustness, we performed 4 additional runs of OVM-guided search on MATH (totaling 7 runs):
>
> | sample size | repeated sampling | OVM-guided search |
> | :--- | :--- | :--- |
> | 1 | 25.5% ± 0.9% | 41.5% ± 1.5% |
> | 32 | 76.1% ± 0.7% | 60.1% ± 0.4% |
>
> The results remain consistent with our main observation: It outperforms repeated sampling when the sample size is small (e.g., 1), but is surpassed at larger sample sizes (e.g., 32).
>
> 6. **Regarding the validity labeling bias**
>
> We thank the reviewer for this insightful observation regarding the directional bias in our generation-vs-selection analysis. We acknowledge that the rollout-based labeling process can be conservative: a valid partial path might be mislabeled as invalid under the limited completions.
>
> However, we emphasize that this bias leads to an underestimation of selection failures. If a false negative occurs (a valid path is mislabeled as invalid), a failure that should have been attributed to the verifier's inability to select it is instead misattributed to the generator's inability to produce it. Consequently, the actual impact of selection failures is likely even more severe than the results reported in Table 2. This bias further reinforces our core claim that the verifier's selection capability—rather than the generator's capacity—is the primary bottleneck causing scaling flaws. We have revised Appendix 6.3 to acknowledge this directional bias and its implications for our causal analysis, colored in red.
>
> 7. **Regarding the promotion of precision results**
>
> We appreciate the reviewer's suggestion to promote the discussion of precision. We agree that while coverage reveals the fundamental scaling flaws of the search paradigm, precision results provide important practical context. Accordingly, we have moved the precision analysis from the appendix to the main text (see the newly added Section 6, highlighted in red).

---

### Review · Reviewer_HoZX · 2026-06-08

**Summary Of Contributions:**

This paper studies verifier-guided beam search for mathematical reasoning. It shows that such search works well at small scale but can lose its advantage over repeated sampling as the sample or candidate size increases. The paper further attributes this to verifier failures, where valid reasoning paths are misranked and pruned. The main strength is the clear empirical observation across several models, datasets, and verifier types. The main limitation is that the study is mostly focused on math reasoning and standard beam search.

**Additional Comments:**

Overall, this is a clear and useful paper. The main finding is simple but important, and the experiments support it well.

**Audience:**

Yes

**Audience Explanation:**

Yes. The findings are relevant to researchers working on LLM reasoning, test-time scaling, verifier models, and search-based decoding. The paper points out a practical limitation of verifier-guided search that is easy to overlook.

**Broader Impact Concerns:**

I do not see major broader impact concerns. A brief note on the computational cost of large-scale inference experiments would be enough.

**Claims And Evidence:**

Yes

**Claims Explanation:**

Yes. The experiments are reasonably broad and consistently support the main claim. The analysis separating generation failures from selection failures also makes the explanation convincing. The evidence is strongest for math reasoning, and the paper should keep the conclusion within this scope.

**Requested Changes:**

1. Please clarify the compute comparison between beam search and repeated sampling, since beam search also requires verifier calls.
2. Please make the conclusion more scoped to mathematical reasoning and standard beam search.
3. Consider moving a short precision discussion from the appendix to the main text.
4. Please briefly discuss the sensitivity to the number of rollouts used to label valid partial paths.
5. Some repeated explanations in Sections 3 and 4 could be shortened.

---

> ### Author Response · Authors · 2026-06-30
> **Rebuttal**
>
> Dear Reviewer HoZX,
>
> We sincerely thank you for your thoughtful evaluation, constructive feedback, and the time dedicated to reviewing our manuscript. We are highly encouraged by your assessment that our main claim—that verifier-guided beam search exhibits a systematic performance reversal relative to repeated sampling—is well-supported by extensive experiments and analysis. We also appreciate your recognition that our work identifies a practical limitation of verifier-guided search that is easily overlooked. We have addressed your insightful critiques in the revised manuscript, with all corresponding updates highlighted in orange.
>
>
>
> 1. **Regarding the concern on the compute comparison between beam search and repeated sampling**
>
> We appreciate the reviewer’s request for clarification on the computational costs. We acknowledge that verifier-guided beam search is computationally more expensive than repeated sampling at the same configuration, as it requires additional verifier forward passes to score candidates at each search step.
>
> However, we emphasize that even with this additional computational budget allocated to the verifier, beam search still fails to outperform—and often underperforms—simple repeated sampling in terms of coverage. This performance gap, despite the higher compute investment in beam search, further underscores the fundamental "scaling flaws" we identified.
>
>
> 2. **Regarding the scope of our research and conclusions**
>
> We appreciate the reviewer’s suggestion to define the scope of our findings more precisely. Accordingly, we have revised the Conclusion section to explicitly frame our insights within the context of mathematical reasoning and standard verifier-guided beam search. Furthermore, we have carefully reviewed the entire manuscript to ensure that our claims regarding "scaling flaws" are consistently attributed to this specific search paradigm and task domain.
>
> 3. **Regarding the promotion of precision discussion**
>
> We appreciate the reviewer's suggestion to promote the discussion of precision to the main text. We agree that while coverage reveals the fundamental scaling flaws of the search paradigm, precision results provide important practical context. Accordingly, we have moved the precision analysis from the appendix to the main text (see the newly added Section 6, highlighted in red).
>
> 4. **Regarding the sensitivity to the number of rollouts for valid path labeling**
>
> We appreciate the reviewer’s suggestion to discuss the sensitivity of rollout numbers. In Appendix A.6.3, we investigate how the number of rollouts impacts the estimation of path validity. Furthermore, we highlight in the revision that the primary conclusion remains robust: selection failure is the dominant bottleneck underlying the scaling flaws, colored in red.

---

### Review · Reviewer_AMwm · 2026-06-17

**Summary Of Contributions:**

(Motivation) LLM performance on mathematical reasoning problems scales with test time compute, using sequential scaling (one long chain of thought) or parallel scaling (multiple candidates). Parallel scaling includes sampling multiple response from the LLM and beam search. Beam search could perform better by using its heuristic to guide search so it is more efficient than repeated sampling. However, beam search performs poorly. This paper measures this and presents investigative experiments to understand it.

(Experiment 1 - Scaling Flaws of Beam Search) Beam search and repeated sampling are evaluated using coverage on GSM8k, MATH, and AIME. Coverage treats a set of samples as successful when at least one answer in the set is correct. Controlling for compute budget, beam search outperforms repeated sampling at small budgets, but always lags behind the performance of repeated sampling at larger budgets. This holds for 3 types of verifiers explored in previous work: Outcome Value Models (OVM), Process Reward Model (PRM), and a Skyworks verifier.

(Experiment 2 - Verifiers cause scaling flaws) A dataset of beam search samples is analyzed to determine whether the poor scaling performance is due to generation or selection. If no valid candidates were generated at the expansion stage of beam search then it is a generation error. If selection removed all valid candidates then it is a selection error and caused by the verifier heuristic. This paper finds ~70% or greater of all the errors are due to selection errors, not generation errors.

(Experiment 3 - Simple verifier alternatives outperform verifiers) When selection is done stochastically based on verifier scores but with high temperature (essentially ignoring those scores) performance is very close to repeated sampling. When selection is done adding final reward (via single rollout) and weighting it higher than verifier score performance also improves modestly.

The conclusion is that verifiers deselect valid solutions during beam search, negating the efficiency gains of beam search at anything but a small amount of compute scaling.

**Audience:**

Yes

**Audience Explanation:**

The practical relevance and general interest of this paper is limited because it uses older open source LLMs for both the mathematical reasoning problem itself and the verifiers. It is also limited because it ignores the sort of reasoning chains used by most modern LLMs.

However it is a well controlled and effective isolation of the cause of performance degradation in these overall less popular and thus less well explored scaling approaches. It will be interesting to people working on test time scaling.

**Claims And Evidence:**

Yes

**Claims Explanation:**

> "verifier-guided beam search suffers scaling flaws: it outperforms repeated sampling at small sample scale but underperforms it at large sample scale."

This is convincingly shown by figure 1, figure 2, and table 1.

> "We pinpoint verifier failures as the primary cause of these flaws."

This is shown by results separating selection from generation in table 2 and figure 4. It is also shown by experiments that interpolate to non-verifier based heuristics in section 5.

> "Our analysis reveals that these issues become more severe for challenging problems, raising concerns about the development of verifier-guided beam search algorithms and their application in real-world settings."

This is less well explored, but basically verified by the AIME result in table 1.

**Requested Changes:**

* > "Qwen3-8B (Yang et al., 2025) is the latest model, but it produces long, sequential chains of thought (CoT) rather than the traditional CoT that our search algorithm operates on."

    It would be useful to add a brief discussion in the main paper that elaborates on the difference and why these experiments cannot be applied to the long sequential thoughts.

* > "For the comparison between beam search and repeated sampling, we align them in terms of “sample size”, which represents the number of complete solution paths generated by each algorithm."

    Could you also compare the computational cost in terms of number of tokens generated?

* The wording "can lead to" in section 4.1 is vague. I think any candidate could lead to a correct final answer. For example, it may be useful for the model to realize a mistake and re-consider. I'm guessing what the authors meant was that a candidate is valid if it is a prefix of a correct complete sample.

---

> ### Author Response · Authors · 2026-06-30
> **Rebuttal**
>
> Dear Reviewer AMwm,
>
> We sincerely thank you for your thoughtful evaluation, constructive feedback, and the time dedicated to reviewing our manuscript. We are highly encouraged by your assessment that our main claim—that verifier-guided beam search exhibits a systematic performance reversal relative to repeated sampling—is convincingly supported. We have addressed your insightful critiques in the revised manuscript, with all corresponding updates highlighted in brown.
>
> 1. **Regarding the fundamental difference between long sequential CoT and traditional CoT in search**
>
> We thank the reviewer for this constructive suggestion. While traditional CoT facilitates exploration by generating multiple parallel sequences, long sequential CoT internalizes the trial-and-error process within a single, extended trajectory. This represents a fundamental distinction in the underlying search mechanism: traditional methods focus on parallel scaling (exploring across multiple paths), whereas long CoT models emphasize sequential scaling (exploring within a single path). Since our verifier-guided beam search operates on step-level transitions and cross-path selection, it is inherently designed for the former and cannot be directly applied to the monolithic internal reasoning of long CoT models. We have added this distinction as a footnote in the model selection paragraph.
>
>
> 2. **Regarding the rationale behind comparison setting**
>
> We deliberately compare verifier-guided beam search and repeated sampling by aligning the sample size, rather than comparing tokens generated, for the two reasons: (1) this setup enables a fair comparison at the selection level (2) this setup further underscores the limitations of verifier-guided search. When aligning the sample size, verifier-guided beam search actually requires more computation than repeated sampling, as it samples more (i.e. $K$) candidates. However, despite the increased computation, its performance is still inferior to repeated sampling. This contrast emphasizes the fundamental limitations of verifier-guided beam search. We have included this discussion in Appendix 4.
>
> 3. **Regarding the definition of valid paths**
>
> We appreciate the reviewer’s request for clarification. The definition of a "valid path" is inherently tied to the underlying search paradigm. In our study, which focuses on parallel scaling using traditional CoT models, the search process is externalized across multiple independent paths. Unlike long sequential CoT (i.e., sequential scaling), traditional CoT models lack the internalized self-correction mechanisms required to recover from an erroneous reasoning step once committed to a trajectory. Therefore, in the context of our paper, the definition of valid paths that "A candidate is regarded as valid if it can lead to the correct final answer" is reasonable.